# Appressoria Formation in Phytopathogenic Fungi Suppressed by Antimicrobial Peptides and Hybrid Peptides from Black Soldier Flies

**DOI:** 10.3390/genes14051096

**Published:** 2023-05-17

**Authors:** Qianlong Sun, Xin Zhang, Ying Ouyang, Pingzhong Yu, Yilong Man, Sheng Guo, Sizhen Liu, Yue Chen, Yunsheng Wang, Xinqiu Tan

**Affiliations:** 1College of Plant Protection, Hunan Agricultural University, Changsha 410128, China; 2Institute of Plant Protection, Hunan Academy of Agricultural Sciences, Changsha 410125, China; 3Longping Branch, College of Biology, Hunan University, Changsha 410125, China; 4College of Plant Science, Hunan Biological and Electromechanical Polytechnic, Changsha 410127, China; 5Institute of Plant Protection, Beijing Academy of Agriculture and Forestry Science, Beijing 100097, China; 6Agricultural Biotechnology Institute, Hunan Academy of Agricultural Sciences, Changsha 410125, China

**Keywords:** AMPs, hybrid peptide, *Colletotrichum acutatum*, *Magnaporthe oryzae*, *Hermetia illucens*

## Abstract

Antimicrobial peptides (AMPs) from black solider flies (*Hermetia illucens*, BSF) exhibiting broad-spectrum antimicrobial activity are the most promising green substitutes for preventing the infection of phytopathogenic fungi; therefore, AMPs have been a focal topic of research. Recently, many studies have focused on the antibacterial activities of BSF AMPs against animal pathogens; however, currently, their antifungal activities against phytopathogenic fungi remain unclear. In this study, 7 AMPs selected from 34 predicted AMPs based on BSF metagenomics were artificially synthesized. When conidia from the hemibiotrophic phytopathogenic fungi *Magnaporthe oryzae* and *Colletotrichum acutatum* were treated with the selected AMPs, three selected AMPs—CAD1, CAD5, and CAD7—showed high appressorium formation inhibited by lengthened germ tubes. Additionally, the MIC_50_ concentrations of the inhibited appressorium formations were 40 μM, 43 μM, and 43 μM for M. oryzae, while 51 μM, 49 μM, and 44 μM were observed for *C. acutatum*, respectively. A tandem hybrid AMP named CAD-Con comprising CAD1, CAD5, and CAD7 significantly enhanced antifungal activities, and the MIC_50_ concentrations against *M. oryzae* and *C. acutatum* were 15 μM and 22 μM, respectively. In comparison with the wild type, they were both significantly reduced in terms of virulence when infection assays were performed using the treated conidia of *M. oryzae* or *C. acutatum* by CAD1, CAD5, CAD7, or CAD-Con. Meanwhile, their expression levels of CAD1, CAD5, and CAD7 could also be activated and significantly increased after the BSF larvae were treated with the conidia of *M. oryzae* or *C. acutatum*, respectively. To our knowledge, the antifungal activities of BSF AMPs against plant pathogenic fungi, which help us to seek potential AMPs with antifungal activities, provide proof of the effectiveness of green control strategies for crop production.

## 1. Introduction

Antimicrobial peptides (AMPs) are small molecules consisting of 10–50 amino acids [1]. AMPs are independent and highly conserved in plants, insects, microorganisms, and mammals [2]. AMPs also exhibit broad-spectrum antibacterial activities, and they are the most promising drug candidates for the prevention and treatment of diseases caused by pathogenic microorganisms [3]. Concerning the development of AMPs, they are a focus of research because of the multitude of their broad-spectrum biological activity functions, low toxicity, low drug resistance in target cells, and ability to regulate host immune responses. Two currently used, different AMPs from plants (Cy-AMP2) and insects (Navidefensin2-2) have been confirmed to exhibit antibacterial activities, but attention should be focused upon whether or not these AMPs have antifungal activity. However, some AMPs from plants (alfAFP, Rs-AFP1, Tk-AMP-D1, Tk-AMP-D1, and Ec-AMP-D1) have been reported to exhibit antifungal activity, mainly due to the activation of these AMPs by corresponding pathogens [4,5,6,7,8]. Interestingly, AMPs from insects (*Hyalophora cecropia*) have been confirmed to exhibit antifungal activity by defensin or cecropin A expression in transgenic plants [9,10,11,12,13]. Moreover, compared to original single peptides, hybrid peptides that cross the active regions of two AMPs together have been confirmed to broaden the antibacterial-spectrum in transgenic plants [14,15]. However, the direct interactions between target AMPs and fungi are still unknown, as are the broad-spectrum antifungal activities.

Insects are a vast resource bank for obtaining ideal AMPs. Since the 1980s when Boman et al. discovered the first AMPs, 339 insect AMPs have been included in the AMP database [16]. AMPs have been confirmed to oppose various animal pathogenic microorganisms, including bacteria, fungi, and viruses [17,18,19]. The black soldier fly (BSF) is a vital insect resource for AMPs found worldwide [20]. The BSF exhibits a defense mechanism against pathogenic microbial infection related to the diverse microorganisms in its living environment and the close contact between the species and pathogenic microorganisms at all growth stages, from larvae to adults. Since there are reports that AMPs from black soldier flies (BSFs) exhibit antibacterial effects, many studies on AMPs have focused on the pathogens that cause animal diseases [21] or regulated microorganism ecosystems [22,23,24]. However, previous studies on BSF AMPs have focused on medicine, food, animal husbandry, and other areas [21], and it is still not clear whether AMPs from BSF can be used as antifungal medicine. Currently, only a few studies have reported the prevention and control effect on plant pathogens when black soldier fly AMPs were used. It is, therefore, expected that this type of cross-kingdom activity can be used against anti-phytopathogenic fungi that cause important diseases in agricultural crops.

The hemibiotrophic phytopathogenic fungi *M. oryzae* causes rice blast, and *C. acutatum* causes anthracnose in peppers. These are important diseases in rice and pepper production, respectively [25]. These two diseases are spread after infection by conidia. Once the conditions are suitable, conidia germinate and form highly specialized infection structures, known as appressoria, on leaf surfaces [26]. Appressoria breach the host’s cuticle and gain access to underlying epidermal cells [27]. Appressoria have varying morphologies that range from undifferentiated germ tube swelling to discrete dome-shaped cells that are separated from the germ tube’s tip by septa [28]. In addition to facilitating plant invasion, appressoria can act as sites for effector delivery and thus mediate the molecular host–pathogen interaction [27]. As a result, appressoria are the ideal targets for evaluating the interactions between AMPs and phytopathogenic fungi conidia, and it is advantageous to observe their inhibitory effects.

Currently, rice blast and pepper anthracnose are controlled mainly by fungicides. However, fungicide overuse has severe consequences for humans and ecosystems, and it easily leads to the development of fungicide-resistant strains [29]. Therefore, developing alternative methods to control crop pathogens is critical. In biological controls, ecological phenomena and some biological characteristics of organism interactions in an ecosystem are used to control plant pests and diseases in addition to microbial pesticides, which are common tools [30]. Searching for antimicrobial proteins has become a popular research topic, and the majority of these proteins with broad antibacterial properties are antimicrobial peptides (AMPs).

Thus, AMPs that are originally from insects have diverse sources, and their potential use in agriculture should be explored [31,32,33]. Therefore, in comparison with single peptides, hybrid peptides that cross the active regions of two AMPs together have been confirmed to broaden the antibacterial spectrum in transgenic plants [14,15]. Here, we attempt to construct an interactive system using BSF AMPs and plant pathogenic fungi and assess selected BSF AMP antifungal activities against two plant hemibiotrophic pathogenic fungi (*M. oryzae* and *C. acutatum*). We found three AMPs—CAD1, CAD5, and CAD7—that showed significant antifungal activity via the inhibition of appressoria formation with respect to the targeted conidia. A tandem hybrid peptide, CAD-Con, was also enhanced with antifungal activity in order to perform comparisons with the single peptides CAD1, CAD5, and CAD7 and the lowered MIC50 concentration in this study. Moreover, pathogenicity was significantly reduced when inoculated with treated conidia on host plants. For single BSF AMPs, active and enhanced expression levels in BSF larvae were observed when treatment was carried out using the conidia of *M. oryzae* and *C. acutatum*, respectively. To our knowledge, the antifungal activities of AMPs from BSF against plant pathogenic fungi assist with the procurement of potential AMPs for controlling plant fungal diseases and provide support for using green production strategies for crop protection in agriculture.

## 2. Materials and Methods

### 2.1. Bioinformatics Analysis of AMPs from the BSF

Whole genome sequences and annotation proteins of the BSF were downloaded from the NCBI RefSeq database (accession no. GCF_905115235.1). Using the reported cecropin amino acid sequences as BlastP query sequences, we identified cecropin-like proteins with E-values of <0.1. Then, the candidates were manually examined by searching the Nr protein database using BlastP [34]. The signal peptide sequences were predicted using SignalP 5.0 with default parameters, and mature cecropin sequences were obtained by cutting off the signal peptide sequences. A neighbor joining tree was constructed based on the multiple alignments of mature sequences by using the neighbor joining method with 1000 bootstrapping replicates. The homologous sequences of other insects were searched in the NCBI database for alignment. The amino acid sequence alignment and phylogenetic tree construction were performed using MEGA-7 software.

### 2.2. Rearing of the BSF Larvae

The BSF larvae were reared with food waste at the Biology and Control of Plant Diseases and Insect Pests Hunan Provincial Key Laboratory, Hunan Agricultural University, Changsha, China. The optimal artificial rearing conditions were as follows: temperature, 28 ± 1 °C; relative humidity, 60–70%; and light/night photoperiod, 12 h/12 h in a greenhouse [35].

### 2.3. Artificial Synthesis of the Screened CAD Peptides

Based on the mature cecropin amino acid sequences, we analyzed the differentiation of 34 predicted candidates from the BSF genome. The seven mature CAD peptides (Appendix A) that were selected in this study were synthesized by Sangon Biotech (Shanghai, China) using Fmoc solid-phase synthesis. The mass spectrometry results are shown in Appendix A. The products were purified to over 90% without any salts. All artificially synthesized peptides were dissolved in ddH_2_O to 500 μM and stored at −80 °C.

### 2.4. Antifungal Activity Assays on M. oryzae and C. acutatum

To evaluate the antifungal activities of the seven selected artificial peptides, the tested fungi were activated on a potato dextrose agar (Solarbio) for 3 days; then, fresh mycelia were treated with working concentrations of 0, 32, 64, 80, and 96 μM of the targeted peptides on hydrophobic plastic dishes. After 24 h, the treated mycelia were observed under a microscope (Axio Observer Zessi, Jena, Germany) [36]. The conidia collected from *C. acutatum* (HHDL02) were isolated from the cultures on CM (complete medium) liquid media [37] while the obtained conidia of *M. oryzae* (Guy11) were washed and isolated on SDC solid media. All aforementioned conidia were centrifuged at 5000 rpm for 10 min at 4 °C and washed twice with sterilized ddH_2_O. The conidia were resuspended at 2.0 × 10^6^ mL^−1^ in ddH_2_O. Then, 12 μL of the resuspended conidia were treated with working concentrations of 0, 32, 64, 80, and 96 μM of the targeted peptides on glass slides. After 2, 4, and 8 h, the conidia germination and appressoria formation were observed under a microscope, and pictures and data were taken and recorded, respectively. Three biological replicates were maintained, with each replicate containing at least 100 conidia. All experiments were repeated three times. Probit regression was used to calculate the 50% minimum inhibitory concentrations (MIC50) of the tested peptides, which were required to inhibit half-spore appressoria formation.

### 2.5. Expression Pattern Analysis of the Selected Target Peptides by RT-qPCR

A total of 0.2 g of sample from the second to sixth BSF larvae were used. Each sample was sterilized with 75% alcohol and washed with DEPC water three times. The insect samples were frozen in liquid nitrogen. Then, using RNAase-free mortars and pestles (Beijing, China), the samples were crushed, and a powder was obtained. The total RNA was extracted and purified using TRIzol [38] (Ambion, Austin, TX, USA) according to the manufacturer’s instructions. The RNA quality and concentration were determined using NanoDrop (BioRad, Hercules, California, USA) after dissolving the RNAs in DEPC water, and then, 20 μL of RNA aliquot was stored at −80 °C.

To confirm the differences in the expression levels of the different target peptides in the BSF larvae, using 1 μL of 10 μM primers of oligosaccharide deoxyribonucleic acid and 1 mg of total RNA, first-strand cDNA was synthesized using a one-step reverse transcription kit (cDNA Synthesis Supermix, TransGen Biotech, Beijing, China). Relative expression levels from the second to sixth target peptides were detected via RT-qPCR. Actin was used as a reference gene. The qPCR was performed using TransStart Green qPCR SuperMix (TransGen Biotech, Beijing, China) under the following conditions: 35 cycles at 94 °C for 5 min, 59 °C for 30 s and 94 °C for 30 s, 72 °C for 30 s, and 72 °C for 10 min. All primers used in the experiments are listed in Appendix A. The data were analyzed in qPCR soft 4.0 software, and the threshold was automatically set by the software of the qTower3G instrument (Analytic jena, Jena, Germany). The expression levels of all tested genes were quantified using the Livak method (relative expression ratio = 2^−ΔΔCT^). 

### 2.6. Cloning and Expression Pattern Analysis of the High-Antifungal-Activity Target Peptides

To clone the four peptides with high antifungal activity, *CAD1, CAD2, CAD5,* and *CAD7* from the BSF cDNA were used as templates. PCR was performed in a 25 μL volume by using a Phanta^®^ Max Super-Fidelity DNA Polymerase Kit (Vazyme, Nanjing, China) and specific primers. The reaction conditions were 35 cycles at 94 °C for 5 min, 59 °C for 30 s, 94 °C for 30 s, 72 °C for 30 s, and 72 °C for 10 min. The fragments from the positive transformants were sequenced and confirmed through alignment using DNAMAN5.0. 

We analyzed the expression pattern of the antifungal peptides from BSF with high activities after infection with *C. acutatum* and *M. oryzae* conidia. Acupuncture treatment was necessary for successful infections, and 200 BSF fourth-age larvae were soaked in 50 mL of *C. acutatum* and *M. oryzae* conidium (2 × 10^8^ mL^−1^) solution for 10 min, respectively. Two positive controls were treated with 3.0 × 10^12^ mL^−1^ of *E. coli* and *Staphylococcus aureus*, respectively, which could activate the AMPs expression in the BSF larvae. Healthy BSF larvae and acupuncture-only-treated larvae were used as the two independent negative controls. All positive and negative control samples were soaked in 50 mL of sterile water for 1 min [39]. All the treated larvae were collected, and the total RNA was extracted using TRIzol. Then, the expression patterns of the four target peptides were detected through RT-qPCR.

### 2.7. CAD-Con Chimeric Vector Construction and Expression in E. coli

Three high-antifungal-activity and high-expression mature peptides, *CAD1, CAD5,* and *CAD7*, were synthesized in a tandem manner and cloned into the T1 vector by TsingKe Co., Ltd. (Nan Jing, China), and the amino acid sequences are shown in Appendix A. The chimeric fragment was subcloned and analyzed via PCR. The purified chimeric fragment was digested with *BamHI/XhoI* and inserted into the pET-SUMO expression vector. The pET-SUMO-CAD-Con recombination vectors were successfully constructed, transformed into *E. coli* (BL21), and grown in LB media containing 50 μg/mL of kanamycin.

The bacterial solution of OD_600_ (approximately 0.6) was obtained through incubation at 37 °C at 200 rpm. Then, 100 μL of 0.5 mM IPTG (Isopropyl β-d-Thiogalactoside) was added to induce CAD-Con expression at 28 °C for 8 h in a 200 mL flask. After centrifugation at 10,000 rpm at 4 °C for 10 min, the cells were collected and resuspended in 3 mL of buffer (1× PBS, 1 mM PMSF). The resuspended cells were ultrasonically disrupted for 10 min on ice. Total proteins were isolated and collected via centrifugation at 10,000 rpm at 4 °C for 10 min. Then, the immunoblotting assay was performed. The total protein was treated using a Bradford Protein Assay Kit (Beyotime), and then, SDS-PAGE was applied [40]. The isolated gel blots were transferred to PVDF membranes and incubated with 5% skim milk powder in 1× TBS-T (*v*/*v*, 0.1% Tween-20) in a shaker for 1 h at room temperature. The blocked membranes were immunoblotted with anti-His antibody (diluted 1:3000 TBS-T) and incubated at room temperature for 1 h. The membranes were washed three times with a 1× TBS-T buffer, diluted with the secondary antibody (goat anti-rabbit LG, 1:5000 G-HRP in TBS-T), and incubated at room temperature for 1 h. The film was washed with a chromogenic solution. The photos were taken using an instrument that scanned the film. At the same time, the other supernatant of the bacteria lysate was purified via affinity chromatography by using a 6× His-tagged Ni column, and the purified target protein was used to assay the antifungal activity of the target protein.

### 2.8. Pathogenicity Assays on Pepper and Rice

To test the differences in the pathogenicities of *C. acutatum* and *M. oryzae,* all tested conidia were collected by centrifugation at 4 °C at 5000 rpm for 10 min, and the precipitates were washed twice with 1 mL of ddH_2_O. Then, the collected conidia were resuspended at 2 × 10^6^ mL^−1^ with ddH_2_O. The conidia of *C. acutatum* were treated with 80 μM of peptides (CAD1, CAD5, CAD7, or CAD-Con) for 1 h at 25 ± 1 °C while the conidia of *M. oryzae* were treated at 28 ± 1 °C. Then, 10 μL of the treated conidia of *C. acutatum* were inoculated on the leaves of the peppers while 10 μL of the treated conidia of *M. oryzae* were inoculated on rice leaves. All inoculated leaves or plants were placed under natural light in a moist incubator at 26 ± 1 °C for 4 days when symptoms appeared. The wild-type conidia treated only with water were used as the control [41]. This experiment was performed three times, with three replicates for each target peptide. All data were recorded for further analysis.

### 2.9. Statistical Analysis

Statistical analyses were performed using SPSS version 21. All collected data with a single variable were analyzed via one-way analysis of variance, and mean separations were performed using Duncan’s multiple range test. Differences at *p* < 0.05 were considered significant [42].

## 3. Results

### 3.1. Amino Acid Sequence Analysis of Cecropin-like Proteins as AMPs from BSF

In total, 34 cecropin-like proteins were identified from the BSF genome annotation, and the amino acids of pre-cecropin were 60–75 nucleotides long. The genes appeared to have been duplicated multiple times in the BSF genome, and 32 of the 34 cecropin-encoding genes were located within a cluster in chromosome five (NC_051853.1) (Appendix A). Similarly, all 34 cecropin-like genes had two exons in their gene structures. All 34 cecropin-like proteins had a single N-terminal peptide. The mature cecropin peptide sequences from the BSF comprised 37–51 aa (average: 46 aa), which were longer than the reported cecropins A and B (37 and 35 aa, respectively). The differences were mainly in the C-terminals, with the common characteristics of amidated residues observed in nature. We chose seven cecropin-like mature peptides for our research based on the amino acid sequence differentiation from the 34 candidates for the subsequent function validation (Figure 1A).

The phylogenetic tree was constructed with the natural enemies of 12 *Diptera*, 7 *Lepidoptera,* and 1 *Coleoptera* from the NCBI database. These seven mature peptides showed 65% similarity with the amino acid sequence of *Dipteran* defensins, but they exhibited 80% similarity with *H. cecropia* and the swallowtail butterfly. The average p distance between *Diptera* and *Lepidoptera* was 1.14 (Figure 1B).

### 3.2. CAD1, CAD5, and CAD7 with Antifungal Activity by the Suppression of Appressorium Formation in C. acutatum and M. oryzae

Seven selected mature peptides were artificially synthesized with C-terminal amidated and observed using mass spectrometry (Appendix A). We found that all seven selected peptides did not affect the hyphal growth of *C. acutatum* when using the working concentration. However, the conidia from *C. acutatum* were subjected to treatment with the seven peptides, and we observed that appressorium formations were significantly inhibited when the conidia were treated with CAD1, CAD5, and CAD7 (Figure 2 and Table 1) while the other four peptides did not work. Moreover, the MIC_50_ concentrations of CAD1, CAD5, and CAD7 were 52 μM, 49 μM, and 44 μM, respectively (Table 1). In practice, we also observed that appressorium formations were significantly inhibited when the conidia of *M. oryzae* were used with CAD1, CAD5, and CAD7, and the MIC_50_ concentrations were 40 μM, 43 μM, and 43 μM, respectively (Table 1). 

In comparison with the wild-type conidia, there was a significant difference observed: the germ tubes were lengthened when both conidia were treated with these three peptides for *C. acutatum* and *M. oryzae*. The average germ tube lengths for *C. acutatum* were up to 289 μm after 2 h of peptide treatments while the average germ tube lengths were up to 793 μm after 4 h of peptide treatments (Table 2). Similar results were observed for *M. oryzae* after 8 h of peptide treatments. 

### 3.3. Expression Patterns of the Selected Peptides from BSF Larvae at Different Instar Ages and under Different Treatment Conditions

Five peptides—*CAD1, CAD2*, *CAD3*, *CAD5*, and *CAD7*—showed significant expression level changes from the second to sixth BSF larvae ages. Their expression levels were lower in the second BSF larvae age than at the remaining time points; then, the fourth larvae age displayed peak expression, and this decreased from the fourth to the sixth larvae ages. Interesting, in the third instar larvae, *CAD1, CAD2, CAD5*, and *CAD7* were significantly upregulated whereas *CAD3* expression was lowered (Figure 3A, *F*_24,50_ = 192.014, *p* < 0.05). Among the five peptides from the fourth larvae age, the expression levels of *CAD7* were up to 64.8 times higher than those in the second instar. The *CAD4* and *CAD6* mRNA expressions were not detected in the experiments, and their expression levels were speculated to be too low to meet the detection limit of the RT-qPCR. 

According to the aforementioned results, *CAD1*, *CAD2*, *CAD5*, and *CAD7* were selected for further investigation of their expression patterns in the fourth instar after inoculation with the conidia of *C. acutatum* and *M. oryzae*. Compared with healthy BSF larvae, no matter what acupuncture treatment was used (the positive control or the treated conidia of *C. acutatum* or *M. oryzae*), their expression levels were significantly enhanced after 24 h. Naturally, the expression levels of these four peptides were also promoted by the conidia from *C. acutatum* or *M. oryzae* compared with independent acupuncture treatments (Figure 3B, *F*_23,48_ = 66.853, *p* < 0.05). Furthermore, the *CAD1*, *CAD5*, and *CAD7* transcripts accumulated at higher levels than *CAD2*, which was consistent with their higher antifungal activities.

To further evaluate the temporal expression patterns, the total mRNA was extracted from the BSF larvae at 12, 24, 36, and 48 h post-inoculation (hpi). All four peptides (*CAD1, CAD2, CAD5*, and *CAD7)* reached their highest expression levels at 24 hpi with conidia from *C. acutatum* or *M. oryzae*, followed by a gradual decline (Figure 3C, *F*_15,32_ = 68.003, *p* < 0.05).

### 3.4. Prokaryotic Expression of the Hybrid Peptide CAD-Con

Although CAD1, CAD5, and CAD7 exhibited antifungal activities, a single protein was too small to meet the prokaryotic expression requirements. Thus, a tandem peptide that included *CAD1, CAD5,* and *CAD7* was constructed, and the recombinant plasmid pET-SUMO-CAD-Con was generated, which was transformed into *E. coli* BL21 (DE3). The CAD-Con fusion protein was observed with an expected band of 27.1 kDa with two tags (SUMO and His) on 12% Tricine-SDS-PAGE (Figure 4A and Appendix A). Additionally, the purified fusion protein was also verified with specificity using Western blotting with the anti-His antibody (Figure 4B).

### 3.5. Hybrid Peptide CAD-Con Had Stronger Antifungal Activity

As expected, CAD-Con also exhibited biological activities against *C. acutatum* and *M. oryzae*. Moreover, the purified CAD-Con protein exhibited better inhibition effects relative to appressoria formations than those of single CAD1, CAD5, and CAD7, and the MIC_50_ concentrations of CAD-Con for *C. acutatum* and *M. oryzae* were 22 μM and 15 μM, respectively (Table 1). To contrast with CAD1, CAD5, and CAD7, under the same concentration treatments, hybrid peptide CAD-Con exhibited 2 and 2.4 times higher inhibitory effects on the appressoria formations for *C. acutatum* and *M. oryzae*, respectively. At the same time, the final germination tube length was approximately 1.3 times longer (Figure 2). However, the inhibitory effect of CAD-Con on the mycelia of plant fungi was not obvious.

### 3.6. Pathogenicity Was Reduced by the Conidia of C. acutatum and M. oryzae Treated with CAD1, CAD5, CAD7, or CAD-Con

The influence of pathogenicity was determined by observing the symptom expressions caused by *C. acutatum* when the conidia were treated with CAD1, CAD5, CAD7, and CAD-Con as inoculums. We found that symptom development was slower on the pepper leaves treated with CAD1, CAD5, CAD7, and CAD-Con whereas typical symptoms were observed on the control leaves at 5 dpi (Figure 5A). Moreover, the mean lesion diameter on inoculated pepper leaves was 3.67 mm, which was significantly (*p* < 0.05) smaller than those of the controls (8.68 mm) (Figure 5A, *F*_4,10_ = 260.725, *p* < 0.05). These results suggested that CAD1, CAD5, CAD7, and CAD-Con could limit disease expression after plant infection with *C. acutatum*. Similar results were observed when rice leaves were used, with treated *M. oryzae* conidia used as the inoculum (Figure 5B, *F*_4,10_ = 56.525, *p* < 0.05).

## 4. Discussion

BSF larvae live in an extreme environment, and this environment is a crucial factor that activates the innate immune systems of BSFs [43]. Their immune system helps this insect species adapt to environmental changes, thereby saving them from extinction [44]. Sericin is a small-molecule AMP synthesized as a secreted protein [43]. It has broad-spectrum resistance to bacteria and a certain inhibitory effect on some fungi affecting humans, such as α-helical (D-V13K and P18), extended (indolicidin), β-sheet (defensins), and Pom-1 [45,46,47,48]. We analyzed the amino acid sequences of all cecropin AMPs. Seven AMPs were screened and cloned. The seven AMPS could be divided into two allelic variant groups; the first one comprised CAD1, CAD3, CAD6, and CAD7 whereas the second comprised CAD2, CAD4, and CAD5. When the amino acid sequences were compared with the homologs, N-terminal conservation was found to be the most common feature, implying that the conserved helical structure was essential for chrysin activity [49]. C-terminal amidation is a common post-translational modification of cecropin, and it may be caused by the presence of C-terminal glycine in enzymatic amidation [50]. Amidation is crucial for the interaction of cecropin with the fat body and contributes to various antimicrobial activities [51]. Whether or not C-terminal amidation is required for the antifungal activity of selected peptides requires further study.

AMPs, an antimicrobial effector secreted by BSF larvae, are required for the adaptation of BSF to their living environment [52]. Most studies have focused on AMPs from older BSF larvae, the AMPs were extracted without examining the instar at which AMP expression was at its peak. We investigated AMP expression in BSF larvae at different ages and under different treatments. We found that the gene expression of the AMP CAD1–7 exhibited a fluctuating pattern in the larval stages; however, all selected AMPs exhibited the highest expression in the fourth instar. Thus, fourth instar BSF larvae were considered more sensitive to outer environmental stress than other instar larvae. After the fourth instar, the epidermis of the larvae gradually hardened and their sensitivity to environmental stress decreased, and then, the *CAD* expressions exhibited a declining trend [53]. The Toll-receptor family led to the activation of the Toll and IMD pathways in the BSF larvae when phytopathogenic fungal conidia were used for treatment after acupuncture, ultimately leading to the induction of defense elements such as the attacin, cecropin, defensin, and drosocin AMPs [54,55,56]. Compared with the control group, the expressions of *CAD1*, *CAD2*, *CAD3*, *CAD5,* and *CAD7* increased after treatment with the fungal spore solution, and the highest AMP expression level was observed 24 h after inoculation. The results showed that phytopathogenic fungi not only triggered the immune response of BSF larvae but also regulated AMP expression with the passage of time, which was consistent with the transcript profiles of the AMPs after treatment with bacteria [53].

To test the hypothesis that the selected BSF AMPs would exhibit antifungal effects, three BSF AMPs with high expression levels were synthesized, and a new heterozygous AMP was expressed via the prokaryotic fusion of the three amino acids of the BSF AMPs [57]. Then, the antifungal effect of a single AMP was compared with that of a new hybrid peptide AMP, CAD-Con. Here, CAD-Con was obtained from purified recombination proteins after it was expressed in the *E. coli* BL21 (DE3) prokaryotic expression system and transformed with a chimeric construct of pET-SUMO-CAD-Con. Hoelecher et al. discovered that fusion to SUMO does not appreciably affect the activity of AMPs [58]. Compared with SUMO-C-L, cecropin-like antimicrobial peptide KR12AGPWR6, and hybrid peptide CLP [57,59,60,61], our hybrid peptide CAD-Con was also expressed normally and exhibited antifungal activities after purification.

Although the AMPs identified from BSFs have exhibited bactericidal activity, antimicrobial activities against plant pathogenic fungi are rarely reported in studies. In our study, CAD1, CAD5, CAD7, and the hybrid peptide CAD-Con did not inhibit mycelial growth or the conidia germination of *M. oryzae* and *C. acutatum*. When the conidia of *M. oryzae* and *C. acutatum* were treated with appropriate concentrations of the target peptide, appressorium formation was suppressed and germ tube growth was accelerated. According to previous studies, germ tube differentiation and appressorium formation may be partly regulated by the Mst11-Mst7-Pmk1 MAP kinase pathway [62]. When the cAMP-PKA pathway was activated, appressorium formation was also inhibited by *Mosfl1*, but the germ tube did not become diapause [63]. In our study, if CAD1, CAD5, CAD7, and the hybrid peptide CAD-Con activate *Mst11* or *Mosfl1* to regulate appressorium formation, there may be another pathway to explore germ tube growth acceleration. Therefore, the mechanisms related to how germ tubes are elongated and the inhibition of appressorium formation by BSF AMPs are worth explaining in the future. Cui Peng et al. discovered that cecropin could cause rough and nicked cell walls and membranes [64], eventually killing *Candida albicans*. Sun Chao-Qin et al. discovered that peptide C18 could damage the cell wall and alter cell membrane permeability [65], resulting in the attenuation of virulence in *C. albicans*. Furthermore, the peptide triggered mitochondrial dysfunction, leading to cell death. Although the working mechanisms of the AMPs cannot currently be explained clearly, this study has provided some clues for understanding the mechanisms. These peptides exhibited stronger effects on pathogenic fungi. Currently, it is speculated that the mutation of amino acids near the C-terminal of the peptide altered the effect. In addition, we are trying to figure out how the AMPs enter fungi. RNA-Seq analyses of the conidia treated with the AMPs can also be further performed in order to explore the involved molecular mechanisms.

Interestingly, appressorium constructs are important for infecting host plant cells with *M. oryzae* and *C. acutatum,* which produce significant osmotic pressure. In our study, pathogenicity was significantly reduced when the pepper or rice leaves were inoculated with the treated conidia. Our results were similar to those observed for *Bacillus velezensis* ZW10, which could inhibit appressorium formation during *M. oryzae* infection [66]. Of note, studies have shown that chlorella and brown algae also significantly reduce appressorium formation in anthracnose pathogen-caused spots on cucumber leaves [67,68]. According to our knowledge, appressoria as the infection structures play important roles in melanin, and their presence is also a sine qua non condition of pathogenicity [69], which suggests that there is a perplexing relationship between melanin and pathogenicity relative to fungi. Recently, many investigations have found that mutants lacking melanin are apathogenic or exhibit reduced virulence [70,71]. Although the conidia treated with the AMPs in this investigation did not develop appressoria, it is unclear whether melanogenesis was typically active during the infection of the host cells. Therefore, the effect of melanogenesis is worth elucidating using transcriptome analyses after conidia are treated by target AMPs. Thus, CAD1, CAD5, CAD7, and the hybrid peptide CAD-Con are promising for the control of relative fungal diseases affecting agricultural production.

The hybridization of different AMPs has been a successful practice for improving the properties of native AMPs [71,72]. Heterozygous peptide CTP can observably improve bactericidal activity [60]. CLP is a novel hybrid peptide derived from CM4, LL37, and TP5 and exhibits higher antibacterial activity than its parental AMPs [73,74]. Synthetic α-helical AMPs with three repeat units [75]—(FFRR)(3), (LLRR)(3), and (LLKK)(3)—were found to be more inhibitory towards *C. albicans*. In this study, compared with CAD1, CAD5, and CAD7, the MIC_50_ value of the hybrid peptide CAD-Con was halved when suppressing appressoria formation. Moreover, the reduced pathogenicity was also significantly enhanced in the experiments. These results indicated that the hybrid peptide had better activities in inhibiting appressorium formation and reducing the virulence of the pathogens. We hypothesized that the antifungal activities increased relative to the number of repeat units or the changed structures. The related mechanisms require further exploration.

## 5. Conclusions

This study demonstrates the potential of BSF larvae in inhibiting infection caused by *M. oryzae* and *C. acutatum* conidia in plants. The AMP CAD1–7 genes were analyzed via RT-qPCR, and three highly expressed AMPs, namely, CAD1, CAD5, and CAD7, and their hybrid peptide CAD-Con were selected to evaluate their antimicrobial activities. The AMPs and hybrid peptides from the BSF larvae could inhibit the fungal diseases caused by the *M. oryzae* and *C. acutatum* conidia. These results indicated that the AMPs produced from the BSF larvae had a key role in blocking the spore infection of plant fungal diseases. Our results provide a theoretical basis for further studies on the mechanisms of BSF larval AMPs against fungal-disease-infected host plants, and BSF AMPs may have applications in agriculture and disease vector control. The mechanisms affecting appressorium formation will be explored in the future when the conidia of *M. oryzae* and *C. acutatum* are treated by BSF AMPs. Studying the modification of genetic plants using BSF AMPs could be helpful for confirming whether host plants can enhance their resistance to fungal pathogens. The present study provides theoretical support for the subsequent prevention and treatment of fungal diseases.

## Figures and Tables

**Figure 1 genes-14-01096-f001:**
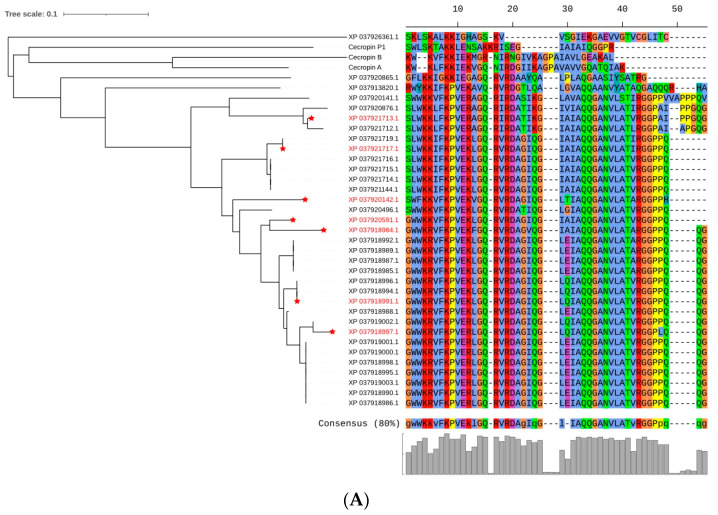
Phylogenetic analysis of the mature peptide sequence of cecropins from the Black Solider Fly (*Hermetia illucens*). (**A**) Multiple alignments of the amino acid sequences of the 34 antimicrobial peptides (marked in red as target peptides). (**B**) Phylogenetic tree of the cecropin families. The amino acid sequences of the cecropins were analyzed using the program MEGA 7.0 with 1000 bootstrap replicates. The number at the nodes indicates the bootstrap proportions (only those above 50% are shown). The 0.1 value represents the branch length, which indicates the relative p distance. The GenBank accession numbers of the insect cecropins are as follows: CAD1 (XP037921713.1); CAD2 (XP037921717.1); CAD3 (XP037918997.1); CAD4 (XP037920141.1); CAD5 (XP037920591.1); CAD6 (XP037918984.1); CAD7 (XP03791899.1); *Drosophila melanogaster* Cecropin C (AAB82509.1); *Drosophila virilis* Cecropin 3 (AAB18324.1); *Drosophila melanogaster* Cecropin A1 (AAF57025.1); *Sarcophaga peregrina* Sarcotoxin IA (AAA29988.1); *Hyalophora cecropia* Cecropin (AAP93872.1); *Agrius convolvuli* Cecropin D (ACX37671.1); *Spodoptera litura* Cecropin D (ABQ51093.1); *Manduca sexta* Cecropin 6 (CAL25128.1); *Bombyx mori* cecropin (AAB30910.1); *Plutella xylostella* cecropin 1 (ADA13281.1); and *Trichoplusia ni* cecropin B (ABV68872.1).

**Figure 2 genes-14-01096-f002:**
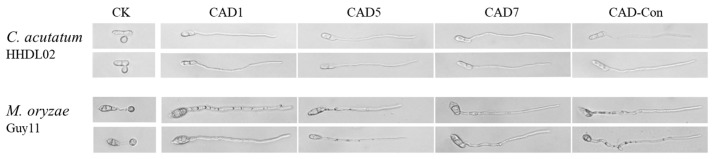
Inhibition of appressorium formations in *C. acutatum* and *M. oryzae* by different peptides. CAD1, CAD5, CAD7, and purified expressed CAD-Con in *E. coli* were added into the conidial suspension. The appressoria were abnormal after treatments with peptides compared with CK. Upper, appressorium formation in *C. acutatum*; lower, appressorium formation in *M. oryzae*.

**Figure 3 genes-14-01096-f003:**
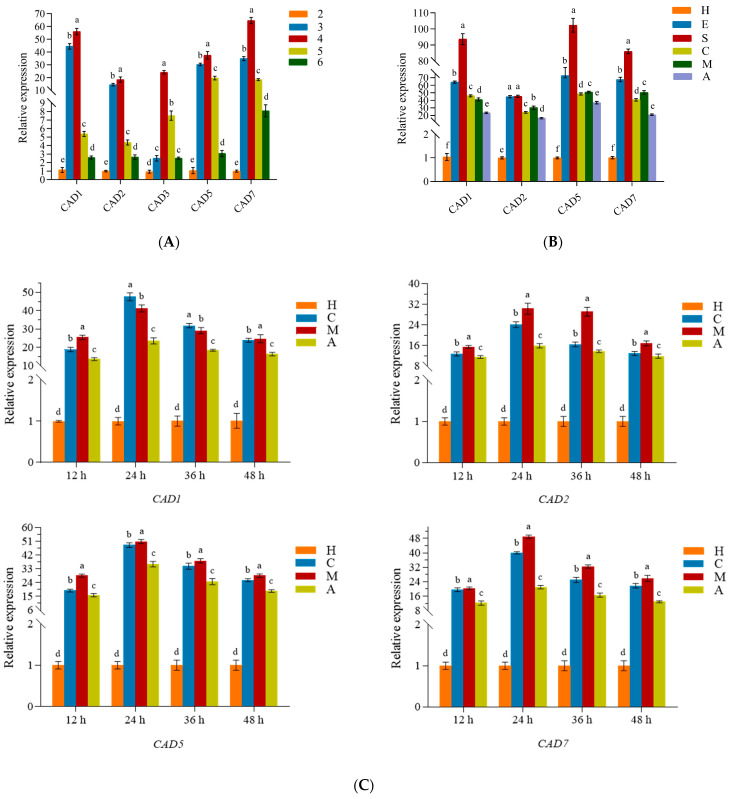
Relative expression levels of the target genes. (**A**) Relative expression levels of the target genes at different insect ages; 2, 3, 4, 5, and 6 indicate the larvae instar. (**B**) Relative expression levels of the target genes after different treatments. H: health; E: *E. coli*; S: *S. aureus*; C: *C. acutatum*; M: *M. oryzae*; A: acupuncture. (**C**) Relative expression levels of the target genes at different time points after fungal treatment. a, b, c, d, e, and f indicate significant differences at *p* < 0.05.

**Figure 4 genes-14-01096-f004:**
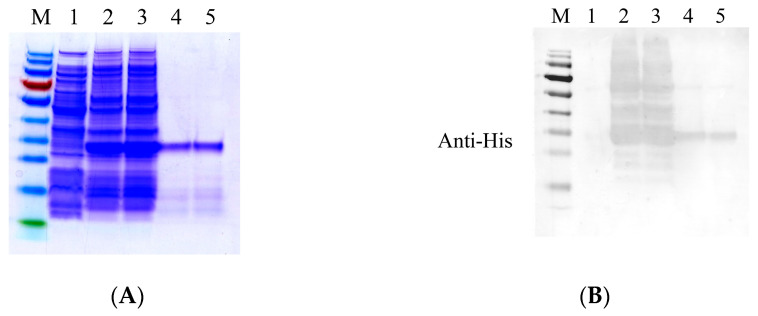
Tricine-SDS-PAGE and Western blot analyses of the CAD-Con expressed in *E. coli.* (**A**,**B**). M: marker; 1: non-induced recombinant protein; 2, 3: IPTG-induced recombinant protein; 4, 5: purified protein.

**Figure 5 genes-14-01096-f005:**
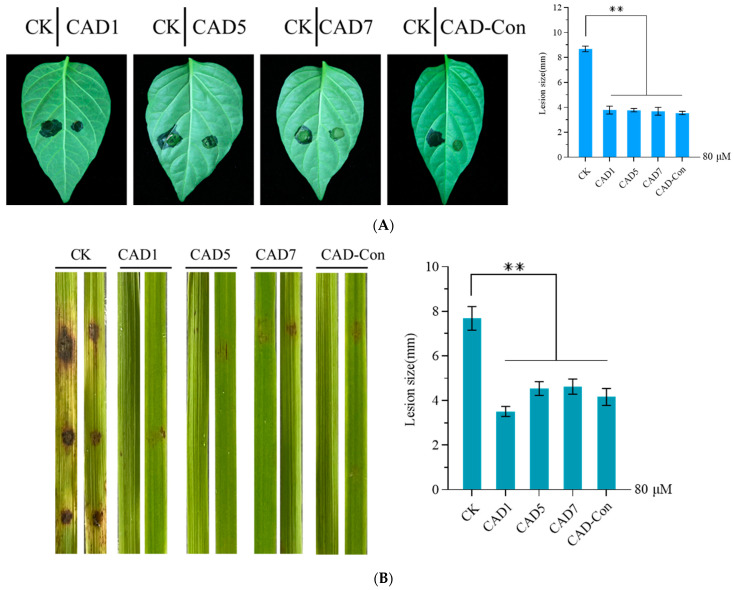
Different target peptides that inhibited infections of the conidia of *C. acutatum.* and *M. oryzae* on host leaves. (**A**) Leaves of three-week-old peppers were inoculated with the conidia of *C. acutatum.* The infected leaves were photographed at 5 dpi (left), and the diameters of the infected areas on the inoculated leaves were measured (right). (**B**) Rice plants at four weeks of age were inoculated with the conidia of *M. oryzae*. The infected leaves were photographed at 7 dpi (left), and the lengths of the infected spots on the inoculated leaves were measured (right). The data show the means ± SDs of three biological replicates. Significant differences were determined by one-way ANOVA analysis (*p* < 0.05), ** means very significant difference.

**Table 1 genes-14-01096-t001:** The MIC50 concentrations of the different peptides for the inhibition of the appressorium formations of *C. acutatum* and *M. oryzae*.

Name	*C. acutatum* (HHDL02) MIC^50^ (μM)	*M. oryzae* (Guy11) MIC^50^ (μM)
CAD1	51	40
CAD5	49	43
CAD7	44	43
CAD-Con	22	15

**Table 2 genes-14-01096-t002:** Comparisons of the germ tube lengths of *C. acutatum* and *M. oryzae* under different treatments.

Strain	*C. acutatum* (HHDL02) Germ Tube (μm)		*M. oryzae* (Guy11) Germ Tube (μm)	
	2 h	4 h	4 h	8 h
CK	6.49 ± 0.50 a	13.50 ± 1.10 a	41.82 ± 8.37 a	74.12 ± 9.44 a
CAD1	293.96 ± 11.40 b	808.30 ± 18.28 b	600.13 ± 16.25 b	1309.88 ± 23.23 b
ACD5	294.81 ± 13.61 b	815.70 ± 16.85 b	633.77 ± 18.57 b	1343.67 ± 15.84 b
ACD7	304.30 ± 14.90 b	799.26 ± 25.03 b	609.12 ± 16.56 b	1322.07 ± 18.61 b
CAD-Con	367.26 ± 19.74 b	986.74 ± 23.73 b	774.09 ± 22.28 b	1613.34 ± 36.75 b

Relative lengths of the germ tubes at different times after peptide treatments. The treatment of *C. acutatum* is given as an example. The values shown are the means ± standard deviations (SDs) of three groups of samples. a indicates a significant difference at *p* < 0.05. The same letter means there was no difference between the experimental group and the control group, and different letters indicate differences between the experimental and control groups.

## Data Availability

The presented data in this study are available upon request from the corresponding author.

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
