# Peer review of "Appressoria Formation in Phytopathogenic Fungi Suppressed by Antimicrobial Peptides and Hybrid Peptides from Black Soldier Flies"

_genes, 2023, doi:10.3390/genes14051096_

Round 1

Reviewer 1 Report

No fungicide was used to compare antifungal activities of AMPs on two important plant pathogens. Without such study we cannot appreciate the  outcome of the research work.

English language was very poor and it was difficult to follow throughout the manuscript.

Author Response

1.No fungicide was used to compare antifungal activities of AMPs on two important plant pathogens. Without such study we cannot appreciate the outcome of the research work.

RESPONSE: It is kind and profitable advice to our future research work, but it is not necessary that fungicide as control to appreciate the effects of target APMs on two fungi in present study. Because screening to obtain target BSF AMPs to work with two important pathogenic fungi, we found that target AMPs were inhibited appressorium formation and promoted germ tube elongation. Besides this aspects, there is reduced in virulence when inoculated with conidia treated by target AMPs. So, our results showed that BSF AMPs have the potentials of control plant diseases caused by fungi in the further. In view of our opinions, your good advices are very helpful to our further research works.

2.English language was very poor and it was difficult to follow throughout the manuscript.

RESPONSE: We carefully check our manuscript and correct some mistakes in grammar, such as abstract part, materials and methods part, results and discussion. At the same time, we also try to improve writing in English. We hope our efforts make your satisfactory.

Reviewer 2 Report

The presented manuscript reports the antimicrobial activities of synthetic peptides from black soldier fly. The manuscript is well written with details, however there are several comments and suggestions as below:

1. The results reported the potential of CAD2 as well, but in the abstract there is not mention of it. Moreover, results section 3.2 did not contain data of CAD2.

2. Is it possible to synthesize the CAD complex containing CAD1, CAD2, CAD5 and CAD7?

3. What is the ATCC or reference number of the microbe strains used in this study?

4. are there any basic characterization data of seven mature CAD peptides ? perhaps can include in the supplementary files?

5. what is the justification of selecting the seven mature CAD peptides out of 34 candidates ?

5. supplementary files are not available for review. Kindly resubmit for revision.

6. at the first glance of the methodology it seems like the authors would sequence the genome of reared BSF instead of obtaining from NCBI database. Please rearrange if the methodology is not as above. 

7. The methodology section 2.5 is not detailed for reproducible. What are the primer sequences ? Table S2 not available for review.

Author Response

1. The results reported the potential of CAD2 as well, but in the abstract there is not mention of it. Moreover, results section 3.2 did not contain data of CAD2.

RESPONSE: According to Figure3 (A), (B) and (C), although the expression patterns of CAD1, CAD2, CAD5 and CAD7 were up-regulated at different instars and under different treatment conditions, the expression levels of CDA2 was lower than that of others. Moreover, 24h after inoculation with conidia of C. acutatum and M. oryzae, the expression peak of CAD1, CAD5 and CAD7 was higher than that of CAD2. In addition, according to figure1(B), Phylogenetic tree showed CAD2 has higher homology with CDA5, however, CAD5 has better response to conidia of C. acutatum and M. oryzae. To takeover, we did not select CAD2 as an important target  to subsequent research.

2. Is it possible to synthesize the CAD complex containing CAD1, CAD2, CAD5 and CAD7?

RESPONSE: We appreciated your helpful and feasible suggestion. In our continuous research, it is worth to anticipation how to make differences by using a tandem hybrid CAD-con of CAD1, CAD2, CAD5 and CAD7.

3. What is the ATCC or reference number of the microbe strains used in this study?

RESPONSE: In this study, the two improtant microbe strains are C. acutatum and M. oryzae. In fact, C. acutatum is C. acutatum HDLL02 strain from our publication article (Chen et al. Simple sequence repeat markers reflect the biological phenotype differentiation and genetic diversity of Colletotrichum gloeosporioides strains from Capsicum annuum L. in China. Journal of Phytopathology. 2021;00:1–9.). M. oryzae is an international common strain oryzae Guy11. Now their detail information was supplemented successfully in the materials and methods part.

4. are there any basic characterization data of seven mature CAD peptides? perhaps can include in the supplementary files?

RESPONSE: The basic characterization data of seven mature CAD peptides was added in the supplementary information, including Number of amino acids, Molecular weight, Theoretical pI and Hydrophilicity (Table S2), these information will be uploaded together with the manuscript. Their synthesis mass spectrum has been included in the supplementary.

5. what is the justification of selecting the seven mature CAD peptides out of 34 candidates?

RESPONSE: The justification of selection target seven mature CAD peptides was based on the differentiation of amino acid sequence from 34 candidates, because all 34 candidates can be divided into seven small groups. In addition, the variation of their length different of selected mature CAD as our consideration factors.

6. supplementary files are not available for review. Kindly resubmit for revision.

RESPONSE: The supplementary files will be uploaded again with our manuscript, please try to open and review again.

7. The first glance of the methodology it seems like the authors would sequence the genome of reared BSF instead of obtaining from NCBI database. Please rearrange if the methodology is not as above.

RESPONSE: We took your kind advice and rearranged the methodology. As your advice, (lines 116-133) the genome was obtained from NCBI database, while the larvae were cultured for RT-qPCR in the study.

8. The methodology section 2.5 is not detailed for reproducible. What are the primer sequences? Table S2 not available for review.

RESPONSE: It is a pity that the supplementary files was unavailable, although Table S3 has been included in the supplementary. We will be uploaded table S3 together with the manuscript again.

Reviewer 3 Report

Good job, but I have some questions that seem important

 The both species Colletotrichum acutatum and Magnaporte oryzae produce appressoria which are key morphological fungal component for infection. In proper work of  appressoria as infection structure important role play melanins, and their presence is a sine quanon condition of pathogenicity. It was confirmed and reported in literature several times that mutants producing non-melanized appressoria are apathogenic. My question is whether applied peptides affected melanogenesis in the target fungi?. The topic should be considered in the discussion.

363 “These results suggested that CAD1, CAD5, CAD7 and CAD-Con could limit mycelia development of C. acutatum in the plants” Authors did not test fungus development inside the plants, the statement was not confirmed by results. It is better to conclude that “ CAD1 …….. could limit disease expression after plant infection with C.a”

It should to be discuss why peptide treatment stimulate growth of germ tube as it was shown in table 2 and on the other hand the same compounds  retarded mycelia development as it was mentioned above.

Author Response

1. The both species Colletotrichum acutatum and Magnaporte oryzae produce appressoria which are key morphological fungal component for infection. In proper work of appressoria as infection structure important role play melanins, and their presence is a sine quanon condition of pathogenicity. It was confirmed and reported in literature several times that mutants producing non-melanized appressoria are apathogenic. My question is whether applied peptides affected melanogenesis in the target fungi. The topic should be considered in the discussion.

RESPONSE: We have made supplement in discussion (line 460-468). According to our knowledge, appressoria as infection structure play important role in melanin, their presence is also a sine qua non condition of pathogenicity[69], which suggested that there is a perplexing relationship between melanin and pathogenicity to fungi. Recently, many investigations have found that mutants lacking melanin are apathogenic or reduced virulence[70,71]. Although treated conidia with AMPs in this investigation did not develop into appressoria, it is unclear if melanogenesis was typically active during infection host cells. Therefore, the effect of melanogenesis is worth to elucidate by transcriptome analysis, after conidia treated by target AMPs.

2. 363 “These results suggested that CAD1, CAD5, CAD7 and CAD-Con could limit mycelia development of C. acutatum in the plants” Authors did not test fungus development inside the plants, the statement was not confirmed by results. It is better to conclude that “CAD1 …….. could limit disease expression after plant infection with C.a”

RESPONSE: It is good advice to infer our concludes. (lines 370-371) we revised “These results suggested that CAD1, CAD5, CAD7 and CAD-Con could limit disease expression after plant infection with C. acutatum.”

3.It should to be discuss why peptide treatment stimulate growth of germ tube as it was shown in table 2 and on the other hand the same compounds retarded mycelia development as it was mentioned above.

RESPONSE: The supplement discussion has been done in the manuscript (lines 434-441). According to previous studies, germ tube differentiation and appressorium formation may be partly regulated by Mst11-Mst7-Pmk1 MAP kinase pathway[62]. When cAMP-PKA pathway is activated, the appressorium formation was also be inhibited by Mosfl1, but the germ tube will not be diapause[63]. In our study, if CAD1, CAD5, CAD7 and the hybrid peptide CAD-Con may activate Mst11 or Mosfl1 to regulate appressorium formation, it is possible that there is another pathway to explore germ tube growth acceleration. So the mechanism how germ tube is elongated and appressorium formation is inhibited by BSF AMPs was worth explaining deeply in the future.

Round 2

Reviewer 1 Report

The authors have revised the manuscript.